# Serotypes and Antibiotic Resistance of *Streptococcus pneumoniae* before and after the Introduction of the 13-Valent Pneumococcal Conjugate Vaccine for Adults and Children in a Rural Area in Japan

**DOI:** 10.3390/pathogens12030493

**Published:** 2023-03-21

**Authors:** Takashi Ono, Masahiro Watanabe, Koichi Hashimoto, Yohei Kume, Mina Chishiki, Hisao Okabe, Masatoki Sato, Sakurako Norito, Bin Chang, Mitsuaki Hosoya

**Affiliations:** 1Department of Pediatrics, Minamiaizu Hospital, Minamiaizu 967-0006, Fukushima, Japan; 2Department of Pediatrics, Fukushima Medical University, Fukushima 960-1295, Fukushima, Japan; 3Department of Bacteriology I, National Institute of Infectious Diseases, Shinjuku-ku 162-8640, Tokyo, Japan

**Keywords:** antibiotic resistance, serotypes, *Streptococcus pneumoniae*, vaccine

## Abstract

The increase in non-vaccine serotypes of *Streptococcus pneumoniae* and their multidrug resistance have become an issue following the introduction of the 13-valent pneumococcal conjugate vaccine (PCV13). In this study, we investigated the serotypes and drug resistance of *S. pneumoniae* detected in adult and pediatric outpatients at a hospital in a rural area of Japan between April 2012 and December 2016. Serotypes of the bacterium were identified using the capsular swelling test and multiplex polymerase chain reaction testing of DNA extracted from the specimens. Antimicrobial susceptibility was determined using the broth microdilution method. The serotype 15A was classified using multilocus sequence typing. The results showed that the prevalence of non-vaccine serotypes increased significantly in children from 50.0% in 2012–2013 to 74.1% in 2016 (*p* ≤ 0.006) and in adults from 15.8% in 2012–2013 to 61.5% in 2016 (*p* ≤ 0.026), but no increase in drug-resistant isolates was evident. However, an increase in the drug-resistant serotypes 15A and 35B was observed in children. Although isolates of these two serotypes showed cefotaxime susceptibility, cefotaxime resistance was confirmed for the serotype 15A isolates. Future trends in the spread of these isolates should be monitored with caution.

## 1. Introduction

*Streptococcus pneumoniae* is a gram-positive, lancet-shaped bacterium that colonizes the human nasopharynx and causes pneumonia, bacteremia, meningitis, and other infections, mainly in children and older individuals [1]. The capsule of *S. pneumoniae*, composed of polysaccharides, is associated with virulence; based on the composition of the capsule, *S. pneumoniae* is classified into more than 100 serotypes [2]. The 7-valent pneumococcal conjugate vaccine (PCV7) developed for children showed a high coverage of pneumococcal serotypes of invasive pneumococcal diseases (IPD) in young children [3]. In Japan, public funding of the PCV7 vaccination for children under five years of age began in November 2010 and the vaccine was incorporated into the routine vaccination schedule in April 2013. It was subsequently replaced with the 13-valent pneumococcal conjugate vaccine (PCV13) in November 2013. In adults aged 65 years and older, the 23-valent pneumococcal polysaccharide vaccine (PPSV23) became a routine vaccination in November 2014.

Several studies have shown a significant decrease in vaccine serotypes (VT) in children as a result of routine vaccination [4,5,6]. A reduction in VT in adults has also been reported [7,8,9,10]. With these serotype changes, the increase in antibiotic-resistant non-VT has become an issue. In particular, an increase in the meropenem-resistant serotype 15A has been reported in Japan [11]. Whether the trends of these changes in serotype and drug susceptibility occur equivalently in remote areas with small populations or whether they have regional characteristics requires clarification. It would also be interesting to confirm the association with the serotypes contained in the 20-valent pneumococcal conjugate vaccine (PCV20), which was approved in the United States (U.S.) in 2021 [12,13]. This new vaccine, which has not yet been approved in Japan, contains seven serotypes (8, 10A, 11A, 12F, 15B, 22F, and 33F) not present in the PCV13. In this study, we investigated changes in *S. pneumoniae* serotypes and antibiotic susceptibility between April 2012 and December 2016 in the Minamiaizu region of Fukushima Prefecture in Japan, which has a declining birthrate and aging population. These results may provide useful information for future pneumococcal vaccine policy.

## 2. Materials and Methods

### 2.1. Study Area and Population

This study was carried out at the Minamiaizu Hospital, located in the Minamiaizu region in the southwest of Fukushima Prefecture in Japan, covering a vast area (about 2340 km^2^). The region had a small population of approximately 27,000 individuals in 2015 and is characterized by a low birthrate and an aging population.

### 2.2. Patients and Bacterial Isolates

This study included 281 *S. pneumoniae* isolates collected from outpatients who visited the Minamiaizu Hospital between April 2012 and March 2013 before the PCV13 routine immunization, between January and December 2014 after the PCV13 routine immunization, and between January and December 2016. Specimens were cultured on a 5% sheep blood agar medium for 18–24 h at 37 °C in 5% carbon dioxide. Bacteria were identified as *S. pneumoniae* if they showed resistance to optochin in the optochin test and were bile soluble in the bile solubility tests against colonies showing α-hemolysis on the blood agar medium. A total of 281 *S. pneumoniae* isolates were collected (Table 1). These clinical isolates were recovered from 233/281 (82.9%) nasopharyngeal swabs, 3/281 (1.1%) otorrhea samples, and 45/281 (16.0%) sputum samples.

Specimens were obtained from patients with physician-diagnosed upper respiratory tract infections, acute bronchitis, acute otitis media, and acute pneumonia. Upper respiratory tract infections were defined as acute pharyngitis, acute tonsillitis, acute sinusitis, and acute rhinitis. The identified diseases belonged to one of the above-mentioned four categories. There were no complication cases, and invasive pneumococcal infection, such as sepsis or meningitis, was not observed in this study. Most of these specimens provided the carrier status of *S. pneumoniae* and did not necessarily indicate the causative pathogen of the disease. The age distribution of the patients according to the clinical isolates were as follows: 44/281 (15.7%) were ≤ 1 year; 170/281 (60.5%) were between 1–4 years; 22/281 (7.8%) belonged to the 5–17 years age group; 2/281 (0.7%) were aged 18–49 years; 7/281 (2.5%) were between 50–64 years; 19/281 (6.8%) were aged 65–84 years; and 17/281 (6.0%) were aged ≥ 85 years. For the purposes of this study, children were defined as those under 18 years of age and adults were defined as those 18 years of age or older. This study was conducted in accordance with the Declaration of Helsinki after obtaining approval from the Ethics Committees of Fukushima Prefectural Minamiaizu Hospital and Fukushima Medical University. This study was approved by the institutional review boards of all the participating institutions. We disclosed information about this study in lieu of written patient consent. The information was disclosed by posting a document describing the significance, purpose, and methods of this study, along with contact information for inquiries regarding this study on the Fukushima Medical University website. The document also clarified that patients who did not wish to participate in this study could consult the contact person in this regard.

### 2.3. Serotyping and Antimicrobial Susceptibility of S. pneumoniae

*Streptococcus pneumoniae* identified from the specimens was stored at −80 °C with Microbank beads (Iwaki & Co., Tokyo, Japan) in the clinical laboratory. DNA was extracted from *S. pneumoniae* using the QuickGene DNA tissue kit S (Kurabo, Osaka, Japan) and was subjected to multiplex polymerase chain reaction (PCR) testing using the U.S. scheme of the Centers for Disease Control and Prevention (CDC) protocol (https://www.cdc.gov/streplab/pneumococcus/resources.html, accessed on 30 January 2023). A primer pair targeting the capsular polysaccharide production gene *cpsA* (160 bp) was used as a positive control in each reaction. All PCRs were performed in a 25 μL reaction mixture. The compositions of each reaction solution and primer sequences were as described in the abovementioned protocol. The PCR conditions were: 95 °C for 15 min, 35 cycles at 94 °C for 30 s, 54 °C for 90 s, and 72 °C for 60 s, followed by a final extension step at 72 °C for 10 min. The products were electrophoresed in 2% Agarose S (NIPPON GENE, Tokyo, Japan) with ethidium bromide at 100 V for 30 min and serotyped according to the size of the products. Finally, serotypes were confirmed with a capsular swelling test using antiserum from the Statens Serum Institut (Copenhagen, Denmark). *Streptococcus pneumoniae* that did not react with any antiserum, had no capsules visible in India ink under the electron microscope, and were confirmed positive for the autolysin enzyme (lytA) gene were defined as non-typeable (NT).

The minimum inhibitory concentration (MIC) values for antibiotic susceptibility testing were determined using the MicroScan autoScan4 (Siemens AG, Munich, Germany) and confirmed using the broth microdilution method described by the Clinical and Laboratory Standards Institute (CLSI) guideline [14]. The following antimicrobials were tested: penicillin, cefotaxime, meropenem, erythromycin, and levofloxacin. *Streptococcus pneumoniae* ATCC49619 was used for quality control. MIC breakpoints were defined according to the CLSI criteria [14].

### 2.4. Multilocus Sequence Typing (MLST)

MLST was conducted using seven housekeeping genes (*aroE*, *gdh*, *gki*, *recP, spi, xpt, ddl*), as described elsewhere [15]. The sequence type (ST) was determined only for *S. pneumoniae* isolates with the serotype 15A. For the 2012–2013, 2014, and 2016 seasons, 6, 6, and 12 specimens were used, respectively. The ST was determined according to the PubMLST database (https://pubmlst.org/spneumoniae/, accessed on 30 January 2023).

### 2.5. Statistical Analysis

The serotype prevalence and antimicrobial resistance rates of *S. pneumoniae* were compared using a Chi-square or Fisher’s exact test based on periods of collection. The statistical significance was set at *p* < 0.05. All analyses were performed using IBM SPSS Statistics version 27 (SPSS Inc., Chicago, IL, USA).

## 3. Results

Of the total 281 isolates, 111 were from 2012–2013, 103 from 2014, and 67 from 2016; 56.2% were male. Most of the study population belonged to the 1–4 years age group (60.5%), while only 3.2% belonged to the 18–64 age group. It was noted that approximately 12.8% of the study participants were older than 65 years. Nasopharyngeal swabs accounted for most (82.9%) of the specimens, while otorrhea accounted for only 1.1%. Moreover, sputum was detected in 16.0% of the study population with all the individuals aged ≥18 and older, of whom the majority were older adults. Acute bronchitis was the most common diagnosis (37.7%), followed by upper respiratory infection (29.5%) and acute pneumonia (22.4%). Acute otitis media (10.3%) was the least common.

### 3.1. Distribution of Serotypes

A significant decrease in the isolates of serotypes included in the PCV13 and an increase in the isolates of serotypes not included in the PCV13 were observed in children (*p* ≤ 0.001). There was also a significant increase in all the isolates of serotypes not included in vaccines (*p* ≤ 0.006) (Table 2). In the 2012–2013 season, 6C was the serotype with the highest prevalence, whereas in the 2016 season, 35B was the most common serotype, followed by 15A (Figure 1).

The PCV20 non-PCV13 serotypes cover seven serotypes (8, 10A, 11A, 12F, 15B, 22F, and 33F). The PPSV 23 non-PCV20 serotypes cover four serotypes (2, 9N, 17F, and 20). PCV13, 13-valent pneumococcal conjugate vaccine; PCV20, 20-valent pneumococcal conjugate vaccine; PPSV23, 23-valent pneumococcal polysaccharide vaccine.

In adults, a significant increase in isolates with serotypes not included in any vaccine was observed (*p* ≤ 0.026) (Table 3); serotype 3 was the most common serotype detected in the 2012–2013 season, while 35B was the most detected in the 2016 season (Figure 2).

The PCV20 non-PCV13 serotypes cover seven serotypes (8, 10A, 11A, 12F, 15B, 22F, and 33F). The PPSV 23 non-PCV20 serotypes cover four serotypes (2, 9N, 17F and 20). PCV13, 13-valent pneumococcal conjugate vaccine; PCV20, 20-valent pneumococcal conjugate vaccine; PPSV23, 23-valent pneumococcal polysaccharide vaccine.

### 3.2. Antimicrobial Susceptibilities

In children, the penicillin and meropenem resistance rates decreased significantly during the 2014 season compared to those during the 2012–2013 and 2016 seasons. In adults, the penicillin resistance rate decreased significantly during the 2016 season compared to that during the 2012–2013 and 2014 seasons (Table 4 and Table 5).

Few resistant isolates of cefotaxime and levofloxacin were observed in children and adults in any of the seasons and more than 80% of isolates were resistant to erythromycin.

In this study, the most frequently detected serotype 6C drug resistance in children during the 2012–2013 season was penicillin resistance in 3.6% (1/28) and meropenem resistance in 0.0% (0/28). We detected a large number of serotype 15A (penicillin resistant: 90.0% [9/10]; meropenem resistant: 60.0% [6/10]) and 35B (penicillin resistant: 64.3% [9/14]; meropenem resistant: 50.0% [7/14]) cases resistant to penicillin and meropenem in children in the 2016 season (Table 6). In adults, serotype 3, the most common in the 2012–2013 season, and serotype 35B, the most common in 2016, were both susceptible in penicillin and meropenem (Table 7).

### 3.3. MLST

All isolates of serotype 15A (6, 6, and 12 isolates in the 2012–2013, 2014, and 2016 seasons, respectively) were classified as ST63.

## 4. Discussion

*Streptococcus pneumoniae* is a bacterium that is endemic to the respiratory tract. Although carrier status does not cause symptoms, the disease develops when the mucosal barrier fails and the bacteria invade the body and multiply. In addition to giving rise to IPD, such as meningitis and bacteremia, potentially life-threatening *S. pneumoniae* is the most common cause of pneumonia and otitis media. Since pneumococcal infections are particularly prevalent in infants and young children, vaccination at an early age is imperative to prevent disease. In addition, since infants and toddlers often congregate in groups at daycare centers and other places of gathering, transmission of *S. pneumoniae* may occur from one child to another, who then brings it home and exposes the adults living in the same household, thereby spreading the disease. IPD in adults and children decreased with the introduction of the PCV7 as a routine immunization in the U.S. in 2000 [16]. In 2010, U.S. routine vaccination substituted the PCV7 with the PCV13 and affected serotype substitution of IPD in adults in 2012–2013 [17]. In Japan, the PCV7 became a routine vaccination for infants in April 2013, and the PCV13 in November 2013; the prevalence of the PCV7 serotypes in adult IPD decreased from 43.3% to 23.8% during 2010–2013 after the introduction of the PCV7 in children [7]. In addition, studies of serotypes in Japanese children and adults with IPD from 2010–2017 showed that the proportion of the PCV13 serotypes decreased from 89.0% to 12.1% in children and from 74.1% to 36.2% in adults [9], indicating an indirect effect of pneumococcal vaccination in children and adults.

This study investigated serotypes and drug susceptibility of *S. pneumoniae* isolates among children and older individuals at a single institution in a remote area of Japan with a low birthrate and an aging population before the introduction of the PCV13 in 2012–2013 and after its introduction in 2014 and 2016. The results showed a significant decrease in the PCV13 serotypes in children after the routine PCV13 vaccination (Table 2) and a decreasing trend in adults, although no significant difference was observed (Table 3). In children, serotype 6C was the most common serotype in the 2012–2013 season, but the serotypes 15A and 35B were detected in large numbers in the 2016 season (Figure 1). In adults, serotype 3 was the most common serotype in the 2012–2013 season, while serotype 35B was the most frequently detected in the 2016 season (Figure 2). Penicillin and meropenem drug resistance decreased significantly in children in the 2014 season, but resistance to both drugs increased in the 2016 season (Table 4). In adults, resistance to penicillin significantly decreased in the 2016 season (Table 5); penicillin and meropenem resistance rates were high for the serotypes 15A and 35B, which were detected in large numbers in children in the 2016 season (Table 6).

Serotype 6C was differentiated from the classic serotype 6A in 2007 [18]. Since the introduction of the PCV7, which contains a serotype 6B conjugate, serotype substitution has occurred, and a number of 6C isolates have been found in other countries [19,20,21]. In Japan, serotype 6C was also detected in large numbers around 2012 [22]. The PCV13, which contains a serotype 6A conjugate, may have cross-protection to serotype 6C [23,24,25]. In this study, serotype 6C was the most common serotype in children in the 2012–2013 season but, in the 2016 season, serotype 6C was drastically reduced to one isolate (Figure 1). This result may be due to the cross-protection of the PCV13. Most of the serotype 6C in children detected in the 2012–2013 season was susceptible to penicillin and meropenem (Table 6). Serotype 6C was detected in a study of serotype trends in IPD and non-IPD in Japan, but no significant increase or drug resistance was observed in this isolate [6,9].

Serotype 3 differs from many other serotypes of *S. pneumoniae* in that it forms mucoid colonies and has high capsule production, resulting in high virulence and resistance to vaccine-induced antibodies [26]. A survey of serotype changes in adults with IPD in 2010–2013 in Japan showed that they accounted for 15.9% (114/715) of the cases and were most frequently detected among cases of acute pneumonia at 22.4% (84/375) [7]. In a nationwide study of IPD in Japan in 2016–2018, serotype 3 was also common, with 15.3% (27/177), along with 12F at 16.4% (29/177) [27]. Thus, the frequency has not decreased in Japan before or after the introduction of the PCV13. In this study, serotype 3 was the most common serotype during the 2012–2013 season in adults (Figure 2). All the adult cases of serotype 3 were also acute pneumonia, although the number of adult specimens was small and most of the specimens were sputum. The nationwide surveillance of pediatric patients in Japan after the introduction of the PCV13 showed a low prevalence rate of serotype 3 among IPD and non-IPD patients in 2014, at 0.8% and 8.5%, respectively [5]. In this study, the proportion of serotype 3 was also low in children in all seasons (Figure 1). Serotype 3 remains one of the predominant serotypes worldwide, even after the introduction of the vaccine, and is more important in older adults [28].

After the introduction of the PCV13, a rapid increase in serotypes 15A and 35B was observed in IPD cases in both adults and children in several countries worldwide [29,30,31,32,33]. Studies in the U.S., U.K., Italy, and other countries have shown that 15A and 35B are endemic in the nasopharynx as well as in IPD cases and are highly resistant to antibiotics [34,35,36,37,38]. In Japan, a significant increase in IPD cases of serotypes 15A and 35B in both adults and children was also observed after the introduction of the PCV7 and the PCV13 [9]. In a nationwide survey of children and adults with and without IPD in Japan between 2015 and 2017, serotype 15A was most frequently detected in non-IPD isolates, followed by 35B. Drug resistance was significantly higher in non-IPD isolates, and serotypes 15A and 35B showed a trend toward multidrug resistance [6]. In a survey of individuals without IPD in Japan in 2018–2019, serotypes 15A and 35B were most frequently detected in children, indicating multidrug resistance [39].

In this study, we observed an increase in serotypes 15A and 35B isolates which were resistant to penicillin and meropenem in children during the 2016 season (Table 6); however, in adults, two serotype 15A and four serotype 35B isolates were detected and susceptible to both penicillin and meropenem in the 2016 season (Table 7). The causes for these observations are unclear, but we may not have detected any resistant isolates in adults due to the limited sample size. As described above, the increase in serotype 15A and 35B drug-resistant *S. pneumoniae* is a problem throughout the world, including Japan. However, the increase in multidrug-resistant *S. pneumoniae* 15A has become a problem in Japan in particular. Nakano S et al. reported an increase in pneumococcal infections caused by meropenem-resistant serotype 15A-ST63 isolates in Japan after the introduction of the PCV13 [11]. In this study, all isolates of serotype 15A were susceptible to cefotaxime and MLST analysis showed all isolates to be ST63. In Japan, serotype 15A ST9084 isolates were detected and resistant to cefotaxime and these isolates were derived from clones of serotype 15A ST63 and generated by a recombination event in the pbp2x region [40]. The future trends of the spread of these isolates should be monitored carefully. In the U.S., the PCV20 was approved for individuals aged 18 and older in June 2021 [12,13]. This new vaccine has not yet been approved in Japan. Seven serotypes (8, 10A, 11A, 12F, 15B, 22F, and 33F) have been added to the PCV20 in addition to the PCV13 serotypes, but serotypes 15A and 35B are not included and there is concern that these serotypes may increase among pneumococcal infections, including IPD.

A limitation of this study is that it is a retrospective study conducted at a single institution. The sample size was small and there were insufficient adult participants to confirm significant differences. Specimens obtained from children may have contained *S. pneumoniae* in a carrier state. Sputum specimens were obtained from all adults, whereas nasopharyngeal swab specimens were obtained from most of the children. Furthermore, the carriage rates of *S. pneumoniae* are particularly high in children compared to those in older adults [38,41,42]. Although vaccination history could not be confirmed, the PCV13 immunization coverage among Japanese children has remained above 95% throughout the study period [9] and the rate was expected to be similar in the regions studied in this study. This study was stopped after 2016 because the physician in charge of this study was transferred to another hospital and after 2019, due to the outbreak of a new coronavirus infection, it became difficult to collect specimens in outpatients and detection of *S. pneumoniae* decreased dramatically. Although we were unable to confirm the serotypes of recent pneumococcal specimens, this study provides important data to Japanese public health officials and bacterial researchers which could help confirm *S. pneumoniae* serotype changes before and after the PCV13.

## 5. Conclusions

Our findings revealed a significant decrease in the pneumococcal serotypes detected in pediatric outpatients after the introduction of the routine PCV13 vaccination in 2016 in remote areas of Japan. In adult outpatients, there was a decreasing trend, although it was not significant. Additionally, we detected a large number of the multidrug-resistant serotypes 15A and 35B in children. Trends in the spread of the multidrug-resistant serotypes 15A and 35B should be noted, as these serotypes are not antagonized by the PCV20 vaccination.

## Figures and Tables

**Figure 1 pathogens-12-00493-f001:**
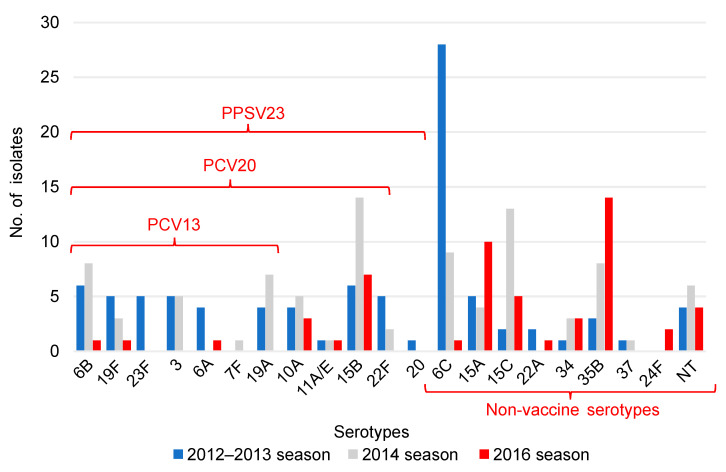
Serotype distribution of *Streptococcus pneumoniae* in children. The vertical axis indicates the number of isolates and the horizontal axis indicates the serotypes of *S. pneumoniae*. Blue represents the number of isolates in 2012, gray in 2014, and red in 2016. PCV13 serotypes dominated in 2012, while non-PCV13 serotypes increased in 2016. However, serotype 6C, which had the highest prevalence in 2012, is classified as a non-vaccine serotype. In 2016, 35B was the most common serotype, followed by 15A. PCV13, the 13-valent pneumococcal conjugate vaccine; PCV20, 20-valent pneumococcal conjugate vaccine; PPSV23, 23-valent pneumococcal polysaccharide vaccine. NT, non-typeable.

**Figure 2 pathogens-12-00493-f002:**
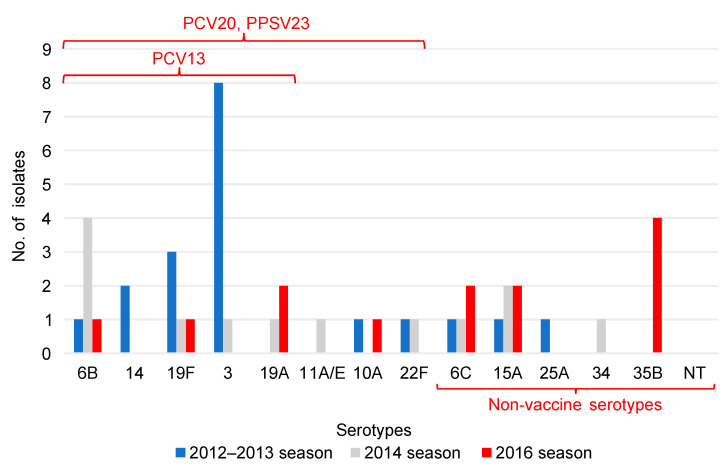
Serotype distribution of *Streptococcus pneumoniae* in adults. The vertical axis indicates the number of isolates and the horizontal axis indicates the serotypes of *S. pneumoniae*. Blue represents the number of isolates in 2012, gray in 2014, and red in 2016. In adults, the number of isolates with vaccine serotypes was high in 2012–2013, but the number of isolates with non-vaccine serotypes increased in 2016. In 2012, serotype 3 was the most common serotype, and in 2016, serotype 35B was the most common. PCV13, the 13-valent pneumococcal conjugate vaccine; PCV20, 20-valent pneumococcal conjugate vaccine; PPSV23, 23-valent pneumococcal polysaccharide vaccine. NT, non-typeable.

**Table 1 pathogens-12-00493-t001:** Characteristics of patients from whom *Streptococcus pneumoniae* isolates were obtained in each study year.

	Number (%) of Isolates			
	2012–2013	2014	2016	Total
No. of isolates	111	103	67	281
Patient characteristics				
Age group				
<1 year	14 (12.6)	16 (15.5)	14 (20.9)	44 (15.7)
1–4 years	66 (59.5)	67 (65.0)	37 (55.2)	170 (60.5)
5–17 years	12 (10.8)	7 (6.8)	3 (4.5)	22 (7.8)
18–49 years	1 (0.9)	0 (0.0)	1 (1.5)	2 (0.7)
50–64 years	6 (5.4)	0 (0.0)	1 (1.5)	7 (2.5)
65–84 years	9 (8.1)	3 (2.9)	7 (10.4)	19 (6.8)
≧85 years	3 (2.7)	10 (9.7)	4 (6.0)	17 (6.0)
Male	64 (57.7)	58 (56.3)	36 (53.7)	158 (56.2)
Specimen type				
Nasopharynx swab	90 (81.1)	90 (87.4)	53 (79.1)	233 (82.9)
Otorrhea	2 (1.8)	0 (0.0)	1 (1.5)	3 (1.1)
Sputum	19 (17.1)	13 (12.6)	13 (19.4)	45 (16.0)
Diagnosis				
Upper respiratory tract infections	28 (25.2)	30 (29.1)	25 (37.3)	83 (29.5)
Acute bronchitis	29 (26.1)	51 (49.5)	26 (38.8)	106 (37.7)
Acute otitis media	25 (22.5)	3 (2.9)	1 (1.5)	29 (10.3)
Acute pneumonia	29 (26.1)	19 (18.4)	15 (22.4)	63 (22.4)

**Table 2 pathogens-12-00493-t002:** Serotype prevalence among isolates of *Streptococcus pneumoniae* from children in each study year.

	Number (%) of Isolates			
	2012–2013	2014	2016	*p* Value
	(*n* = 92)	(*n* = 90)	(*n* = 54)	
PCV13 serotypes	29 (31.5)	24 (26.7)	3 (5.6)	0.001
Non PCV13 serotypes	63 (68.5)	66 (73.3)	51 (94.4)	0.001
PCV20 non-PCV13 serotypes	16 (17.4)	22 (24.4)	11 (20.4)	0.501
PPSV 23 non-PCV20 serotypes	1 (1.1)	0 (0.0)	0 (0.0)	0.456
Total Non-Vaccine serotypes	46 (50.0)	44 (48.9)	40 (74.1)	0.006

**Table 3 pathogens-12-00493-t003:** Serotype prevalence among isolates of *Streptococcus pneumoniae* from adults in each study year.

	Number (%) of Isolates			
	2012–2013	2014	2016	*p* Value
	(*n* = 19)	(*n* = 13)	(*n* = 13)	
PCV13 serotypes	14 (73.7)	7 (53.8)	4 (30.7)	0.056
Non-PCV13 serotypes	5 (26.3)	6 (46.2)	9 (69.2)	0.056
PCV20 non-PCV13 serotypes	2 (10.5)	2 (15.4)	1 (7.7)	0.818
PPSV 23 non-PCV20 serotypes	0 (0.0)	0 (0.0)	0 (0.0)	NA
Total non-vaccine serotypes	3 (15.8)	4 (30.7)	8 (61.5)	0.026

**Table 4 pathogens-12-00493-t004:** Antimicrobial resistance rates among isolates of *Streptococcus pneumoniae* from children in each study year.

	Number (%) of Isolates			
Antimicrobial	2012–2013	2014	2016	*p* Value
	(*n* = 92)	(*n* = 90)	(*n* = 54)	
PCG	25 (27.2)	14 (15.6)	21 (38.9)	0.007
CTX	5 (5.4)	1 (1.1)	0 (0.0)	0.072
MEPM	14 (15.2)	5 (5.6)	15 (27.8)	0.001
EM	83 (90.2)	85 (94.4)	53 (98.1)	0.153
LVFX	2 (2.2)	1 (1.1)	0 (0.0)	0.519

PCG, penicillin; CTX, cefotaxime; MEPM, meropenem; EM, erythromycin; LVFX, levofloxacin.

**Table 5 pathogens-12-00493-t005:** Antimicrobial resistance rates among isolates of *Streptococcus pneumoniae* from adults in each study year.

	Number (%) of Isolates			
Antimicrobial	2012–2013	2014	2016	*p* Value
	(*n* = 19)	(*n* = 13)	(*n* = 13)	
PCG	4 (21.1)	5 (38.5)	0 (0.0)	0.049
CTX	1 (5.3)	0 (0.0)	0 (0.0)	0.497
MEPM	2 (10.5)	3 (23.1)	0 (0.0)	0.172
EM	16 (84.2)	12 (92.3)	11 (84.6)	0.777
LVFX	1 (5.3)	0 (0.0)	0 (0.0)	0.497

PCG, penicillin; CTX, cefotaxime; MEPM, meropenem; EM, erythromycin; LVFX, levofloxacin.

**Table 6 pathogens-12-00493-t006:** Antimicrobial resistance rates of serotypes 3, 6C, 15A, and 35B *Streptococcus pneumoniae* in children in each study year.

	Rates (%) of Resistant Isolates					
	3			6C		
Antimicrobial	2012–2013	2014	2016	2012–2013	2014	2016
	(*n* = 5)	(*n* = 5)	(*n* = 0)	(*n* = 28)	(*n* = 9)	(*n* = 1)
PCG	0.0	0.0		3.6	0.0	0.0
CTX	0.0	0.0		0.0	0.0	0.0
MEPM	0.0	0.0		0.0	0.0	0.0
EM	100.0	100.0		100.0	100.0	100.0
LVFX	0.0	0.0		0.0	0.0	0.0
	**15A**			**35B**		
**Antimicrobial**	**2012–2013**	**2014**	**2016**	**2012–2013**	**2014**	**2016**
	**(*n* = 5)**	**(*n* = 4)**	**(*n* = 10)**	**(*n* = 3)**	**(*n* = 8)**	**(*n* = 14)**
PCG	60.0	25.0	90.0	0.0	0.0	64.3
CTX	0.0	0.0	0.0	0.0	0.0	0.0
MEPM	60.0	25.0	60.0	0.0	0.0	64.3
EM	100.0	100.0	100.0	100.0	100.0	100.0
LVFX	20.0	0.0	0.0	0.0	0.0	0.0

PCG, penicillin; CTX, cefotaxime; MEPM, meropenem; EM, erythromycin; LVFX, levofloxacin.

**Table 7 pathogens-12-00493-t007:** Antimicrobial resistance rates of serotypes 3, 6C, 15A, and 35B *Streptococcus pneumoniae* in adults in each study year.

	Rates (%) of Resistant Isolates					
	3			6C		
Antimicrobial	2012–2013	2014	2016	2012–2013	2014	2016
	(*n* = 8)	(*n* = 1)	(*n* = 0)	(*n* = 1)	(*n* = 1)	(*n* = 2)
PCG	0.0	0.0		0.0	0.0	0.0
CTX	0.0	0.0		0.0	0.0	0.0
MEPM	0.0	0.0		0.0	0.0	0.0
EM	100.0	100.0		100.0	100.0	100.0
LVFX	12.5	0.0		0.0	0.0	0.0
	**15A**			**35B**		
**Antimicrobial**	**2012–2013**	**2014**	**2016**	**2012–2013**	**2014**	**2016**
	**(*n* = 1)**	**(*n* = 2)**	**(*n* = 2)**	**(*n* = 0)**	**(*n* = 0)**	**(*n* = 4)**
PCG	0.0	100.0	0.0			0.0
CTX	0.0	0.0	0.0			0.0
MEPM	0.0	100.0	0.0			0.0
EM	100.0	100.0	50.0			100.0
LVFX	0.0	0.0	0.0			0.0

PCG, penicillin; CTX, cefotaxime; MEPM, meropenem; EM, erythromycin; LVFX, levofloxacin.

## Data Availability

Not applicable.

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
