# Peer review of "Serotypes and Antibiotic Resistance of Streptococcus pneumoniae before and after the Introduction of the 13-Valent Pneumococcal Conjugate Vaccine for Adults and Children in a Rural Area in Japan"

_pathogens, 2023, doi:10.3390/pathogens12030493_

Round 1

Reviewer 1 Report

In this manuscript, the authors describe serotype distribution and antibiotic resistance profiles of S. pneumonia isolates from a rural area in Japan. The authors describe solid data, but the presentation shows several flaws. First, it is not clear if the sampled pneumococci are causative pathogens, or carriage isolates. This should be clearly mentioned and discussed. Second, the description of the antibiotic resistance profile is very confusing, even though the data are good. Authors should describe these findings more clearly. Third, authors should correctly use the words strain, isolate, sample and serotype. The English of the manuscript needs some serious improvement at several points. I advise editing by a native speaker. Even with these remarks, I think the data are good and deserve publication. O thorough work-over will certainly make this publishable.

Major remarks:

Line 52: PCV20 was approved for adults in the US in 2021. This should be added already here. Not only in the discussion.

Line 71ff: Even though your patients were diagnosed with URTI; taking a nasopharyngeal swab will most probably give you the carrier status of the patient, and not necessarily the causative pathogen of the disease (especially for young children). This should be mentioned already here, and not only in the discussion.

Line 208: The children are spreading the bacteria, not necessarily the disease.

Line 218: Prefer to write you investigated the drug susceptibility of s. pneumonia isolates causing infections.

Line 296: Here it is finally mentioned that most of your nasopharyngeal isolates might be carriage. This means that the pneumococci are not necessarily the responsible pathogen. Do not write ‘carriers of the disease’.  This point should be made more clear througout the manuscript. It has to be made clear what you are looking at: carriage, or disease causing pathogen.

Minor remarks:

Line 21: …were identified using multilocus…

Line 23: …non-vaccine serotype…

Line 26:  Although isolates of these two serotypes showed…

Line 27:  ….. for serotype 15A isolates in Japan.

Line 34: The capsule of S. pneumonia… (the capsule is not a membrane…)

Line 39: ..and the vaccine was incorporated….

Line 99: ….were as described in the abovementioned protocol.

Line 114: ..only for S. pneumonia isolates with serotype 15A.

Line 135: …with the highest prevalence; while…..

Line 145:  Serotype prevalence of Streptococcus….

Line 147: …2012-2013….  You cannot say: ‘PCV13 containing strains’ Rephrase this as ‘PCV13 serotypes’ or ‘ isolates with serotypes included in PCV13’. Please correct this throughout the entire text.

Line 153: …. Increase in isolates with serotypes not included in any vaccine ….

Line 165:  …the number of isolates with vaccine serotypes was high in 2012-2013, but the number of isolates with non-vaccine serotypes increased in 2016.

Line 171, and rest of text. Please do not use abbreviation PCG and MEPM in the text, write out in normal words penicillin and meropenem. You can use the abbreviations in the graphs and the tables, but in the text it makes it unreadable.

Line 171: This whole phrase should be reformulated, as this is not proper English. It is completely unclear what is decreasing or increasing. Susceptibility? Resistance? To which comparison does the p-value in tables 4 and 5 relate?

Line 180. Same remark as for line 171.

Line 201: …bacteria invade the body and multiply. (bacteria are plural)

Line 205: …prevent disease.

Line 210:  …substituted PCV7 with PCV13 and…

Line 212: …that the prevalence of PCV7 serotypes in adult…

Line 215: PCV13-included serotypes..

Line 222: see remarks before

Line 230: do not abbreviate

Lines 239-241: This sentence is not readable, please rephrase in proper English.

Line 248: …showed that serotype 3 accounted for… The second part of the sentence does not make sense. Did the study have 715 or 375 participants? If 114/375 had IPD, how can 84/375 have pneumonia? What was the study population?

Line 275: …two serotype 15A and four serotype 35B isolates were detected and all were susceptible…

Line 283: 15A isolates

Lines 284-286: Rephrase in proper English.

Line 287: In the U.S., PCV20 was….

Line 290:  ….. PCV13 serotypes, but…

Line 291: …increase among pneumococcal infections…

Line 296: …from carriage, since…

Line 299: region

Reviewer 2 Report

I underdstand the study is rerospective and data is slightly out dated due to unforseen circumstances and the coronavirus pandemic as stated in the discussion. I can appreciate that the study is intended for  public health and it gives insight into the antibiotic resistance trends of non-vaccine pneumococcus in Japan. 

Could the authors please include full method of the antibiogram, indicating how the antibiogram were read whether this was using a plate reader or by eye.

Can the authors also comment on non-capsulated strains, although predominantly commensals they are also known to causes disease, did the authors come across these strains?

Can the authors also comment on why MLST was only performed against serotype 15A?

Round 2

Reviewer 1 Report

Authors have dealt with all issues raised by the reviewers.

There are only two small remarks on the English wording:

Line 222. omit 'always', as carriage is by definition asymptomatic. That is why it is called asymptomatic carriage.

Line 228just write 'transmission'. Do not use the word 'cross', as it is redundant.
